# AT-Hook Transcription Factors Show Functions in *Liriodendron chinense* under Drought Stress and Somatic Embryogenesis

**DOI:** 10.3390/plants12061353

**Published:** 2023-03-17

**Authors:** Yao Tang, Weihuang Wu, Xueyan Zheng, Lu Lu, Xinying Chen, Zhaodong Hao, Siqin Liu, Ying Chen

**Affiliations:** 1Key Laboratory of Forest Genetics and Biotechnology of Ministry of Education, Co-Innovation Center for the Sustainable Forestry in Southern China, Nanjing Forestry University, Nanjing 210037, China; 2National Germplasm Bank of Chinese fir at Fujian Yangkou Forest Farm, Shunchang 353211, China; 3Key Laboratory of Forest Genetics and Biotechnology of Ministry of Education, Nanjing Forestry University, Nanjing 210037, China

**Keywords:** abiotic stress, AT-hook motif, genome-wide, transcription factor

## Abstract

AT-hook motif nuclear localized (AHL) is a transcription factor that can directly induce plant somatic embryogenesis without adding exogenous hormones. One of its functional domains, the AT-hook motif, has a chromatin-modifying function and participates in various cellular processes, including DNA replication and repair and gene transcription leading to cell growth. *Liriodendron chinense* (Hemsl.) Sargent is an important ornamental and timber tree in China. However, its low drought-resistant ability further leads to a low natural growth rate of its population. Based on bioinformatics analysis, this study identified a total of 21 *LcAHLs* in *L. chinense*. To explore the expression pattern of the *AHL* gene family under drought and somatic embryogenesis, we performed a systematic analysis including basic characteristics, gene structure, chromosome localization, replication event, cis-acting elements and phylogenetic analyses. According to the phylogenetic tree, the 21 *LcAHL* genes are divided into three separate clades (Clade I, II, and III). Cis-acting element analysis indicated the involvement of the *LcAHL* genes in drought, cold, light, and auxin regulation. In the generated drought stress transcriptome, a total of eight *LcAHL* genes showed increased expression levels, with their expression peaking at 3 h and leveling off after 1 d. Nearly all *LcAHL* genes were highly expressed in the process of somatic embryogenesis. In this study, we performed a genome-wide analysis of the *LcAHL* gene family and found that *LcAHLs* take part in resistance to drought stress and the development of somatic embryos. These findings will provide an important theoretical basis for understanding of the *LcAHL* gene function.

## 1. Introduction

The AHL gene family (AT-hook motif nuclear localized gene family) has been studied in a wide variety of plant species, such as *Oryza sativa* L., *Arabidopsis thaliana*, *Gossypium raimondii*, *Zea mays*, *Vitis vinifera*, *Glycine max* L. Merr, *Populus trichocarpa* and others [1,2,3,4,5,6]. For example, 29 *AHL* gene family members were identified in *Arabidopsis*, and 48, 51, and 99 *AHLs* were identified within the *G. Raimondi*, *G. arboretum*, and *G. hirsutum* genomes, respectively [2,3]. Furthermore, 14 *AHLs* were found in grape [4], and 37 *AHLs* were identified in maize [1]. Recently, some reports have shown that *AHL* plays an important role in stress response, especially in abiotic stress [6]. In *P. trichocarpa* and *Oryza sativa* L., the expression of *AHL* was significantly upregulated under drought stress [7]. The abiotic stress-responsive *SOB3/AHL29* transcription factor limits petiole elongation by antagonizing growth and promoting *PIF* [8]. In addition, *AHLs* are related to plant growth and development, as *AtAHL16* plays a role in the transcriptional activation of anther development [9], and *AtAHL18* is involved in regulating root system architecture and growth [10]. Furthermore, some researchers speculated that *AtAHLs* might involve either direct transcriptional regulation of target genes or more global regulation by altering chromosome structure in the process of somatic embryogenesis and polyploid generation [11,12].

*Liriodendron*, belonging to the magnolia family, was previously widely distributed in temperate regions of the northern hemisphere, but numerous species have gone extinct as a consequence of the expanding quaternary glaciers. At present, just two species of *L. chinense* and *L. tulipifera* remain [13]. Hybrids between *Liriodendron chinense* and *Liriodendron tulipiferea* possess a highly productive somatic embryogenesis system [14]. *L. chinense* is an ancient relic plant and one of the nationally rare and endangered protected plants in China. As a precious commercial tree species [15], it has a high growth speed, high-quality wood texture, mandarin jacket shape leaves and calendula-like flowers. Drought stress is the most prevalent environmental factor limiting plant growth and reproduction [16]. However, the adaptability to arid environments of *L. chinense* is weak, resulting in a low natural reproduction rate. Previous studies have shown that *L. chinense* grows in subtropical monsoon climate areas and that precipitation in the warmest quarter exerts the greatest impact on its growth [17]. Under drought stress, its low natural reproduction efficiency and seed germination rate limit regeneration of the natural populations [18]. The embryogenic callus of *L. Hybrid* possessed a mature genetic transformation system [19]. Therefore, finding an alternative gene that improves drought resistance and somatic embryogenesis efficiency can help to cultivate and promote new transgenic varieties with excellent resistance.

Somatic embryogenesis is a type of plant cell totipotency where embryos develop from nonreproductive cells without fertilization [20]. Two main methods for inducing somatic embryogenesis have been reported, which are direct somatic embryogenesis and indirect somatic embryogenesis [21,22]. A suspension culture system of embryogenic calli of the *Liriodendron hybrid* was established with high proliferation after explants successfully regenerated from immature zygotic embryos [14]. Somatic embryogenesis is widely used in plants such as *Arabica coffee* [23], ornamental bananas (*Musa,* spp) [24], and tree peony (*Paeonia* sect. *Moutan*) [25]. In *Arabidopsis*, overexpression of *AHL15* can induce direct somatic embryogenesis without adding exogenous hormones [12]. All these studies have not systematically explored the expression of *AHLs* at different stages of somatic embryogenesis; furthermore, most of the studies on *AHLs* resistance to abiotic stress are focused on model species such as *Arabidopsis thaliana*, and few studies were conducted in woody species. As an important transcription factor and direct downstream target gene of *BBM*, there is no detailed report on the genome-wide identification of the *LcAHL* gene family in *L. chinense* at present. With the publication of the *L. chinense* genome [17,26], we can explore and study its important functional genes [27] to provide candidate genes for the cultivation of transgenic elite species with high somatic embryogenesis efficiency and drought resistance. We identified 21 *LcAHL* genes and analyzed using bioinformatics, including a description of their physical and chemical properties, basic characteristics, gene structure, chromosome localization, replication events, cis-acting elements, protein tertiary structure and phylogenetics. Afterward, our transcriptome analysis showed that the expression of *LcAHL* increased significantly during drought stress and ABA-induced somatic embryogenesis. The evolutionary analysis is helpful in fully understanding the origin and evolution of the *AHL* genes in *L. chinense* and to lay the foundation for its gene function research. Many gene structures and promoter analyses also provide a basis for exploring its gene regulatory network. Targeted research and transformation of genes related to drought stress and somatic embryogenesis in the *L. chinense* genome help to improve the ability to adapt to a drought environment and lay the foundation for further development of quality timber industry. The study also provides candidate genes for the cultivation of transgenic elite species with high somatic embryogenesis efficiency and drought resistance.

## 2. Results

### 2.1. Identification and Physicochemical Property Analysis of LcAHL Gene Family Members

We identified a total of 21 putative full-length genes encoding LcAHL proteins in the *L. chinense* genome (Appendix A). Our analysis of the tertiary structure suggests that the LcAHL proteins are composed of α-helices, β-folds and multiple γ-corners. The proteins are coiled and folded to form a precise and complex tertiary structure (Figure 1a). They possess coincident conserved PPC/DUF296 domains and AT-hook motifs (Figure 1b).

We then analyzed the physical and chemical properties of LcAHLs (Table 1), such as their pI (isoelectric point), MW (molecular weight), instability index and subcellular localization. Their predicted pI values ranged from 4.71 to 10.62; thus, 14 LcAHLs can be classified as basic proteins with theoretical isoelectric points exceeding 7, and the remaining LcAHLs are acidic proteins. Their molecular weight ranged from 22.38 kDa (Lc06945) to 72.82 kDa (Lchi06288), and their grand average of hydropathicity was from −0.551 (Lchi06750) to 0.055 (Lchi01373). The instability index of all LcAHL proteins scored a value over 40; since proteins with an instability index below 40 are generally considered stable [28], we conclude that all the LcAHL proteins are unstable proteins. Subcellular localization predictions indicated that most LcAHL (16/21) proteins localize to the nucleus, which is consistent with previous studies. The remaining LcAHLs are predicted to localize to either chloroplast and/or mitochondria: Lchi04180 localizes to mitochondria, Lchi30606 and Lchi12845 to chloroplasts, Lchi22185 to both mitochondria and chloroplasts and finally Lchi13190 to both the nucleus and chloroplasts.

### 2.2. Phylogenetic Analysis of LcAHL Proteins

The result of multiple sequence alignment and evolutionary tree clustering showed that all 21 LcAHLs, with only Lchi22185 excluded, possess both a PPC/DUF296 domain and AT-hook motif (Figure 2). The AT-hook motif is composed of three amino acids, which is R-G-R. The PCC domain is volatile and can be divided into two types: L-R-S-H and F-T-P-H [29]. Based on the protein sequence of their conserved domains, the 21 LcAHLs could be divided into two clades: Clade A and Clade B. To infer the evolutionary relationship between LcAHLs and AtAHLs, we carried out Bayesian evolutionary analysis using BEAST (v2.2.6) software [30]. A phylogenetic tree was constructed with the full-length AtAHLs (29) and LcAHLs (21), which could be divided into three clades: Clade I, Clade II, and Clade III (Appendix A). Clade A was consistent with Clade I, and Clade B was further subdivided into Clade II and Clade III (Figure 3a).

Previous studies suggested that *L. chinense* is one of the earliest diverging core angiosperms [31]. To further analyze the phylogenetic interrelatedness of the *LcAHL* gene family, the protein sequences of the *AHL* gene family members from 21 species (Appendix A), including 2 *Bryophyta* and *Pteridophyta*, 4 *Monocots* and 15 *Dicots*, were constructed into an evolutionary tree (Figure 3b). In conclusion, the phylogenetic tree showed the relative consistency and diversity of the evolutionary status of AHL in different species. It provided evidence for the homology analysis of AHL among different species.

The species tree indicates that the evolutionary position of the LcAHL proteins lies between *Ricinus communis* and *Carica papaya* (Figure 3b)*,* meaning that *LcAHLs* are most closely related to homologous proteins from dicotyledonous species. However, *Liriodendron* itself is classified as a basic angiosperm based on its full genomic sequence. This discrepancy could be caused by differing speeds of functional differentiation within the *AHL* family. From the species tree analysis, we found that the number of *AHL* family members in dicotyledonous was higher than the number in monocots and basic angiosperms. We infer that the expansion rate of the *LcAHL* gene family is not the same as that of the Liriodendron genome. The number of left-after genome-wide replication events and environmental selection is close to the number of dicots. Most dicots have experienced genome-wide duplication events; thus, the number of *AHLs* is higher than that of monocots and basal angiosperms.

### 2.3. Distribution of the LcAHL Gene Family across the Liriodendron Genome

Based on the available *Liriodendron* genome annotation data, we mapped the chromosome locations of 21 *LcAHLs* (Figure 4 and Appendix A). We found that a total of nineteen *LcAHLs* are unevenly distributed across nine chromosomes, and two *LcAHLs* are located on a scaffold. Four *LcAHL* genes are located on chromosome 11, and only one *LcAHL* is present on chromosomes 4, 7, and 18. Chromosomes 1 and 8 contain three *LcAHLs*, while chromosomes 3, 6, 10 and Scaffold1025 each contain two *LcAHLs*. Gene tandem duplication and fragment duplication events are frequent in the process of gene evolution (Appendix A). Through the analysis of replication events, we found that 14 *LcAHL* genes show signs of fragment replication, while only a single pair of genes is the result of a tandem replication, being *Lchi06945* and *Lchi22185*. Therefore, the results of this study indicated that follows that fragment duplication was the fundamental expansion mechanism of the *LcAHL* gene family, while a small number of genes experienced tandem replication.

### 2.4. LcAHL Gene Structure and Promoter Analysis

We used the MEME website to predict six conserved domain models of LcAHL protein family members, including 11–50 amino acids and 5–20 sites (Figure 5a and Appendix A). Combined with multiple sequence alignment analyses, we found motifs 1 and 2 to be highly conserved. In agreement with previous studies, these two motifs are the core conserved sequences of LcAHLs. Almost all proteins contain motif 1 or motif 2, indicating the high degree of sequence conservation within the LcAHL protein family.

Gene structure analysis (Figure 5b) showed the *LcAHL* gene length and the quantitative relationship between the coding sequence (CDS) and untranslated region (UTR). CDS length ranges from 600 to 1000 bp, genomic sequence length from 904 to 27,410 bp, and LcAHL protein length ranges from 151 aa to 474 aa (Table 2). The introns make up a large proportion of the gene space, and the number of exons varies from 3 to 6. Genes in Clade II and Clade III possess at least two introns more than those in Clade I. Most members of Clade I subgroup did not have UTR, except for *Lchi06288*. Clade II members contain more introns and have a relatively dense and short CDS. Excluding a small number of genes that do not have UTRs, such as *Lchi01372* and *Lchi04864*, other gene members in Clade II possess one to three UTRs. These results indicate the diversity of the *LcAHL* protein sequences.

The *cis*-acting elements within promoters are generally related to transcriptional regulation of gene activity dynamic networks, including abiotic stress responses, hormone responses, and developmental processes [32]. The *cis*-acting elements present within the *LcAHL* promoter include elements that function in stress resistance, such as MBS, which is related to drought resistance, LTR, which has a role in low-temperature stress, and AE-box, MRE, I-box, GT1 motif, GATA motif, and G-box elements, which are related to light response and hormone response elements (Table 3). There are many cis-acting elements of *LcAHL* related to abiotic stress, such as LTR, TC-rich repeats and MBS [33]. Therefore, we hypothesized that the AHL protein may be involved in stress resistance. At the same time, the promoter region of *LcAHLs* also possesses many cis-acting elements related to cell cycle regulation (MSA-like) and endosperm expression [34,35]. Therefore, we speculate that AHL proteins may be involved in early embryonic development.

### 2.5. LcAHLs Expression Analysis during Drought Stress and Somatic Embryogenesis

We performed transcriptome analysis of the *AHL* gene in different tissues of *Liriodendron chinense*. The results showed that *LcAHLs* are mainly expressed in the stigma, bud, stamen, leaf, bark, and sepal (Figure 6a and Appendix A). For example, *Lchi13190* and *Lchi05954* are mainly expressed in reproductive organs, such as the stamen, sigma, and sepal, while their overall expression levels are low in vegetative organs. *Lchi06945* is mainly expressed in the stamen, while *Lchi32451* is mainly expressed in the xylem. If a gene is expressed highly in a specific tissue, this usually indicates that the gene’s function is related to that organ’s function [36]. Through tissue-specific expression information, we found that *LcAHLs’* expression in reproductive organs is significantly higher than that in vegetative organs. Therefore, we speculate that *LcAHL* gene function may be related to plant reproductive growth, such as embryogenesis.

Subsequently, transcriptome analysis of AHL gene under drought stress was also performed. It shows that the expression of *LcAHLs* increased during drought stress (Figure 6b and Appendix A). After one hour of drought stress, different *AHL* genes showed three different expression patterns (Appendix A). Pattern I: gene expression began to increase one hour after drought stress, including *Lchi13190* and *Lchi13244*. Pattern II: Gene expression began to rise after three hours of drought stress and stabilized after one day. Pattern III: the remaining did not respond significantly to drought stress. In conclusion, the gene expression levels of *LcAHLs* increased from 3 to 12 h and thereafter gradually decreased to their previous levels.

In addition, we used qRT-PCR to quantify the expression trends of three *LcAHLs* (*Lchi05954*, *Lchi06280,* and *Lchi13190*) in different seedling tissues after drought stress treatment simulated by 15% PEG6000 (Appendix A). The results showed that the *LcAHL* genes responded immediately to drought stress, especially after 3 h of drought stress. The expression of *LcAHLs* began to decrease after 12 h, consistent with the results of our transcriptome analysis. At the same time, we found that upon 15% PEG6000 treatment, *LcAHL* expression responded earlier in roots and stems than in leaves. For example, *Lchi05954* expression was upregulated after 3 h in roots and stems, while the expression in leaves only began to rise after 72 h (Figure 7).

During the process of somatic embryogenesis, the gene expression of *LcAHLs* increased significantly during the transformation from proembryo to spherical embryo (Figure 8). Pro-embryogenic masses were cultured for 20 days on 3/4 MS medium to generate embryogenic callus (PEMs) and then cultured in liquid suspension for another 10 days, after which single cells were cultured for 2 days. Embryo samples were induced by ABA for increasing time intervals to collect successively staged embryos under the microscope: 1 day for early embryos (ES3), 7 days for globular embryos (ES5), 31 days for mature cotyledon embryos (ES9) and 37 days for plantlets (PL). Transcriptome analysis of somatic embryos shows that *LcAHL* genes are expressed during somatic embryogenesis; we could discriminate two unique *LcAHL* expression patterns during this process. Expression pattern I: after ABA treatment, the expression of *AHLs* increased and then reached the lowest expression at the spherical embryo stage (ES5), finally increasing at the later stages of somatic embryogenesis. Pattern II is lower during the early stage (ES1-4) and then increases during the mid-term (ES5-7) and later stages (ES7-9).

We selected nine *LcAHL* genes to verify the transcriptome data in five critical periods by qRT-PCR (Figure 9a and Appendix A). Somatic embryos were collected at PEM, S3, S5, S9 and PL stages (Figure 9b and Appendix A). We divided the expression patterns into three types. Type I: the expression of *LcAHLs* has no significant difference with PEMs during somatic embryogenesis. However, when induced by ABA (S3), the expression trend decreased. Type II: after ABA induction (S3), the expression of AHL increased (S5), then decreased and finally peaked at the stage of plants (PL). Type III: the expression of *AHLs* increased continuously during somatic embryogenesis.

### 2.6. LcAHL Subcellular Localization

To explore the potential function of *LcAHL* genes in transcriptional regulation, we verified their predicted subcellular localization using *L. chinense* callus protoplast transient transformation (Figure 10). We used a ZEISS LSM 800 fluorescence microscope (Carl Zeiss, Germany) to observe the expression pattern of *LcAHLs* under different light waves. Cellular morphology and certain cellular structures, such as starch granules, vacuoles, etc., can be seen using visible light, while we used GFP fluorescence to visualize GFP-tagged LcAHL proteins. Then, *35S::GFP* was used as a control construct, lighting up the entire cell in GFP fluorescence. The subcellular localization assay showed that the encoded proteins of *Lchi05954* and *Lchi13190* only exhibited fluorescent signal in the nucleus but not in the cytoplasm and cell membrane. Therefore, we speculate that it may play a regulatory role as a nuclear transcription factor during transcription.

## 3. Discussion

### 3.1. Diversity of LcAHL Family Members, Their Gene Structure and Physicochemical Properties

We initially identified 28 *LcAHL* proteins in *L. chinense* that possess a PCC domain. However, after verifying their protein domains using the Pfam website, we found that only 21 proteins possess a bona fide AT-hook motif and PCC domain. Although *Lchi22185* has no conserved domain sequence, it nonetheless can be clustered within the *AtAHL* evolutionary tree. Genome duplications, non-neutral selection, and co-evolving residues may lead to changes in the conserved domain of proteins [37]. Thus, we speculate that *Lchi22185* may have lost its conserved domain during evolution. Previous studies have shown that there are 28 *AHL*s in Arabidopsis, 63 family members in soybean [5], 37 *AHL*s in popular [6], 14 in grape [4], 37 family members in maize [1] and 47 *AHL* genes in carrot [38]. In our study, we divided them into two clades, according to their characteristics and distribution of conserved domains. Based on phylogenetic tree analysis with *A. thaliana* (S 1), we further divided Clade B into two subgroups. We found that the Clade A protein sequences have remained relatively conserved, whereas the Clade B proteins were more strongly diversified during evolution. From a protein level, the PPC domain in Clade A changes greatly, which is consistent with analyses performed in soybean [5]. It may be related to the diversity of *LcAHL* gene function [39].

In soybean, *AHL* genes can reach more than 6 kb in length, with a CDS length of approximately 1 kb [5]. The longest gene sequence in *L. chinense* is over 20 kb, with the average CDS length being similar to soybean [5]. In plants, the number of genes containing introns is significantly higher than in previously reported studies [40]. We found that all 21 *LcAHL*s contain introns, indicating that this feature may have been conserved evolutionarily and could be important functionally. These genes may use introns to create novel splice variants to tune gene function to a specific developmental stage and tissue type [41]. In maize, only one gene in Clade I has a UTR, indicating that the *AHL* gene structure has diversified among different species [1]. A combination of the motif and gene structure analysis showed that Clade II and Clade III evolved from Clade I [5], similar to what has been found for the maize *AHL* gene family [1].

Most LcAHL proteins are neutral to weak basic proteins, similar to cotton AHLs [3], while soybean AHLs are more acidic. Previous studies have shown that high hydrophobicity is not only conducive to the internal folding of proteins but is also important for the nuclear localization of the PPC domain [42]. The amino acid hydrophobicity of LcAHLs can reach 70 kDa, higher than that of soybean, cotton, and grape, indicating that the *LcAHL* proteins are relatively stable.

According to species tree analysis, the evolutionary status of *LcAHLs* is closely related to *Carica papaya* and *Ricinus communis*, which belong to dicotyledons in evolutionary status. Transcription factor genes of the same family but from diverse eukaryotic organisms show structural and functional similarity, suggesting that they evolved from a common ancestor [43]. In prokaryotes and early eukaryotes, the association with AT-hook motifs is not necessary for the function of PPC protein domains [44]. With the occurrence of gene duplication events, the number of *AHL* gene family members increases and gene functions differentiated. Similar reports have been reported in rice. The authors speculate that gene replication may be a key mechanism for the functional diversification of the rice YSL (yellow stripe-Like) gene family, leading to changes in spatial and/or temporal gene expression [45]. In the basal *Bryophyta Pteridophyta* genome, a small number of *AHL*s is present, while in later diverging species, the number of *AHL* genes gradually increased. The gene number variation of the *AHL* gene family may be regulated by several factors. However, natural selection may play a more important role in ginseng [46].

Based on subcellular localization algorithms, LcAHLs were predicted to be localized to the nucleus. Previous studies have shown that the hydrophobic region of the PPC domain is crucial for its nuclear localization. Mutations of the AT-hook motif may lead to the chromatin-binding function being lost [47]. We used homeostatic transformation to transfer *Lchi05954*/13190 into protoplasts and verified these two AHLs to be localized to the nucleus. We further speculate that they may function as transcription factors.

### 3.2. LcAHLs Have Expanded Primarily through Fragment Duplication

Eukaryotic genomes are replicated from large numbers of replication origins distributed across multiple chromosomes [48]. To explore the evolution of *LcAHLs*, we investigated the collinearity between genomes. In our research, we found the *LcAHL* genes to be distributed across nine chromosomes and one scaffold, unrelated to chromosome size and location. We also found that fragment replication has been the dominant form of replication in the *LcAHL* gene family, with a small amount of tandem replication taking place, which contrasted with the most common dispersed replication observed in maize [1]. In cotton, the amplification of the *AHL* gene family has taken place via fragment replication or genome-wide replication, with no tandem replication having taken place [3]. This indicates that the amplification mode of the *AHL* gene family varies across different species. Large-scale replication events (21/15) have occurred in the *LcAHL* gene family, mainly in Clade A. A large number of examples exist where changes in gene activation state follow replication events: genes may be either transcriptionally activated or repressed [49].

### 3.3. LcAHLs Expression Pattern Analysis

In soybeans, *AHL*s are mainly expressed in roots, meristems and epicotyls. In cabbage, *AHL*s are mainly expressed in roots and buds. In maize, they are mainly expressed in roots, embryos, endosperm, and seeds [1]. We analyzed the expression of *LcAHLs* in different *Liriodendron* tissues and found that, excluding a small number of *LcAHL* genes expressed in the phloem, such as *Lchi06945*, *Lchi13190,* and *Lchi05954*, they are mainly expressed in floral organs, which may be related to their biological function. Among them, *Lchi13190* and *Lchi05954* are highly expressed in various floral tissues, while *Lchi06945* is mainly expressed in stamens with low expression in other tissues. *AHL15* and other *AHL* Clade A genes function directly downstream of flowering genes, such as *SOC1* (SUPPRESSOR OF OVEREXPRESSION OF CONSTANS 1), *FUL*(FRUITFULL), and upstream of the flowering-promoting hormone gibberellic acid [50]. Drought is an important stress, having a huge impact on the growth and productivity of plants [51]. In order to resist drought stress, it is urgent to explore drought resistance genes and cultivate high drought-resistant varieties. However, comprehensive phenotyping at the seedling stage is an efficient way to select a drought-tolerant germplasm [52]. Our results suggest that *LcAHLs* increase gene expression after 3 h of drought stress in somatic embryo seedlings. Therefore, we speculate that they may be involved in drought stress resistance.

The *cis*-acting elements are usually closely related to the function of the downstream gene, such as the 150 bp *cis*-acting sequence located upstream of the TATA box of *AtNRT2.1*, which confers regulation of NO^3−^, sugars and N metabolites [53]. Our analysis of *cis*-acting elements present in the promoters of *LcAHL* genes detected many elements related to drought stress. For example, *Lchi01373* and *Lchi04864* possess MYB binding sites that are inducible by drought. Drought is one of the major causes of dramatic yield loss in plants [43]. Plants respond and adapt to water stress by increasing the expression of water-stress-inducible genes that respond to mitigate the effects of water stress [54]. The qRT-PCR results of *LcAHLs* showed that roots respond faster than stems and leaves under drought stress. *Lchi05954* expression gradually increased after 3 h in the root and stem, experiencing a peak after 12 h, after which it gradually declined, while it only increased after a period of 72 h in leaves.

The *AHL* gene family encodes embryophyte-specific nuclear proteins with DNA binding activity [10]. *AtAHL15* and other *AtAHL* genes are necessary for normal embryonic pattern formation and development after the globular stage [12]. From our transcriptome analysis, we found that the expression of *LcAHL* increased during the transformation from proembryo to globular embryo and that the expression of *Lchi04180* and *Lchi12845* increased significantly at the ES5 stage. Previous studies have shown that AHL functions during the decondensation of chromatin during cellular mitosis. Overexpression of AtAHL15 can increase the probability of tetraploid formation during somatic embryo induction [12]. Therefore, exploring *LcAHL* genes could improve the efficiency of *L. chinense* somatic embryogenesis in a meaningful way.

## 4. Materials and Methods

### 4.1. Genome-Wide Identification of LcAHL Proteins

The genome sequence file, annotation file and CDS sequence of *L. chinense* were acquired from NCBI (https://www.ncbi.nlm.nih.gov/, accessed on 5 September 2022), while the 29 AtAHL amino acid sequences were downloaded from TAIR (https://www.arabidopsis.org/, accessed on 5 September 2022). The HMM file (Hidden Markov model file) of the PCC domain was retrieved from the Pfam database (http://pfam.xfam.org/, accessed on 5 September 2022) and used to search for protein sequences containing a PCC domain with the HMMER program (v3.3.6) in the total proteome of *L. chinense*. Next, the protein sequences of the *LcAHL* family were extracted from the *L. chinense* protein sequence database using the identified protein number. BLAST+(2.7.1) was used for homology comparison to 29 *AtAHLs*, and the CDD website (https://www.ncbi.nlm.nih.gov/cdd, accessed on 5 September 2022) and Pfam were used to confirm and delete proteins without a PCC domain. Subsequently, the correct nonredundant family members were obtained.

The LcAHL motif was predicted using the MEME website [55] (https://meme-suite.org/meme/doc/meme.html, accessed on 5 September 2022) at an E-value of 10^−5^ (maximum number of motifs is 6; maximum width is 50; minimum sites is 2; maximum sites is 600), after which an evolutionary tree was constructed using TBtools software (v1.09). The intron/exon gene structure was determined using the gene feature file (gff3) and displayed with the GSDS website (http://gsds.gao-lab.org/, accessed on 5 September 2022). The basic protein physical and chemical properties, such as protein length, molecular weight (MW) and isoelectric point (pI), were determined with the ProtParam tool of the ExPASY website (https://www.expasy.org/, accessed on 5 September 2022). Subcellular localization analysis was performed using the Plant-mPLoc website (http://www.csbio.sjtu.edu.cn/bioinf/plant-multi/, accessed on 5 September 2022). Promoter element analysis was completed using the PlantCARE website (http://bioinformatics.psb.ugent.be/webtools/plantcare/html/, accessed on 5 September 2022).

Protein tertiary structure was predicted using the SWISS-MODEL website (https://swissmodel.expasy.org/interactive, accessed on 5 September 2022). The corresponding PDB file was downloaded, visualizing the tertiary structure of the protein using the Swiss PDB viewer (v4.0.1).

### 4.2. Multiple Sequence Alignment and Evolutionary Analysis

ClustalX2 (v2.1) software was used to construct a multiple sequence alignment of *LcAHL* protein sequences (the gap being set as the default value), which was subsequently visualized on the ESPript website (https://espript.ibcp.fr/ESPript/cgi-bin/ESPript.cgi, accessed on 5 September 2022). The evolutionary tree of *L. chinense* and *A. thaliana* was drawn using RAxMAL (v8.2.11) and then displayed on the iTOL website (https://itol.embl.de/, accessed on 5 September 2022).

AHL protein sequences of 21 species, including *Bryophyta*, *Pteridophyte*, *Monocots,* and *Dicotyledoneae*, were retrieved from the Phytozome website (https://phytozome-next.jgi.doe.gov/, accessed on 5 September 2022) and aligned with ClustalX. Multiple sequence alignment was saved in FASTA format and converted into an XML file with the BEAUti program, selecting Dayhoff for the site model. Finally, we used the TreeAnnotator program to annotate the evolutionary tree, where the burin percentage was set to 90 and the limit to 1. A Bayesian tree was completed and visualized by Figtree software (v1.4.3).

### 4.3. LcAHL Chromosomal Location and Replication Event Analysis

Genome annotation files were used to calculate gene length, chromosome length, and gene chromosomal position. MCScanX software was applied to calculate replication events while setting the parameters as follows: similarity > 70%, e-value cutoff < 1E^−6^. The R software package Circlize [56] was used to visualize the chromosomal location.

### 4.4. Cis-Acting Element Analysis

The 1.5 kb upstream region of *LcAHL* family members was extracted and analyzed using the PlantCARE website (http://bioinformatics.psb.ugent.be/webtools/plantcare/html/, accessed on 5 September 2022), and the *cis*-acting elements were visualized by TBtools software.

### 4.5. LcAHL Expression Pattern Analysis under Drought Stress and Somatic Embryogenesis

The drought stress transcriptome data of *L. hybrid* is annotated with accession number PRJNA679101 and can be downloaded through NCBI (https://www.ncbi.nlm.nih.gov/bioproject/PRJNA679101/, accessed on 5 September 2022). The date of somatic embryogenesis is unpublished. Heatmaps visualizing expressions were displayed with TBtools software (v1.09). The log2 (TPM + 1) value was used for standardization and hierarchical cluster analysis.

Plant materials used were the *L. hybrid* somatic embryo line 166,302, cultivated in the greenhouse for 2 weeks (16 h of light/8 h of darkness and 75% relative humidity). Then, 15% PEG6000 [57] was used to simulate a naturally arid environment, performing three biological replicates. Roots, stems and leaves were harvested after 3 h, 12 h and 3 d drought treatment, immediately frozen with liquid nitrogen and stored at −80 °C. Somatic embryogenesis was induced from immature zygotic embryos. Pro-embryogenic masses were cultured for 20 days on ¾ MS medium to generate embryogenic callus (PEMs) and then cultured in liquid suspension for another 10 days, after which single cells were cultured for 2 days. Embryo samples were induced by ABA for increasing time intervals to collect successively staged embryos under the microscope: 1 day for early embryo’s (ES3), 7 days for globular embryos (ES5), 31 days for mature cotyledon embryo (ES9) and 37 days for plantlets (PL).

Total RNA was extracted using a fast pure total RNA Isolation Kit (enzyme, Nanjing, China) 527 (RC401) and then reverse transcribed into cDNA with a script ^®^ III 1st strand cDNA synthesis 529 Kit (Vazyme, Nanjing, China). CDS sequences were obtained from NCBI (https://www.ncbi.nlm.nih.gov/assembly/GCA_003013855.2. , accessed on 5 September 2022). Primers were designed using Oligo7 (v7.6) with a length from 80 to 200 bp and then blasted against the *L. chinense* genome using TBtools to ensure their specificity (Appendix A). *Lc18S* was used as an internal reference gene. The composition of the amplification mix was 20 μL: 10 μL 2x AceQ ^®^ qPCR SYBR ^®^ Green Master Mix (Without ROX), 0.4 μL specific primers (Forward/reverse primers), 2 μL cDNA template, and 7.2 μL ddH_2_O in a final volume of 20 μL. The reaction procedure was as follows: pre-denaturation at 95 °C for 30 s; denaturation at 95 °C for 10 s; annealing at 60 °C for 30 s; extension at 72 °C for 30 s; with 40 cycles in total, repeated three times. Then, the fluorescence value (CP value) obtained from the reaction result was calculated with the 2^−∆∆CT^ method [58].

### 4.6. LcAHLs Subcellular Localization

We used the pJIT166-GFP vector to construct a *LcAHL*-GFP fusion protein [59]. To obtain the fusion protein expression vectors, *Lchi05954* and *Lchi13190* were recombined with pJIT166-GFP, using the SalI and XbaI restriction enzymes. *L. chinense* callus was hydrolyzed with cellulose, hemicellulose and pectinase for 8 h to prepare protoplasts. The two recombinant plasmids and the empty plasmid pJIT166-GFP as control were then transformed into the prepared protoplasts using a PEG8000 mediated method [60], after which they were cultured in the dark for 24 h. We used a ZEISS LSM 800 fluorescence microscope (Carl Zeiss, Germany) to observe LcAHL subcellular localization.

## 5. Conclusions

In this study, we identified and analyzed 21 LcAHL proteins in *L. chinense*, finding that they have similar protein structures and functional domains. Based on comparison of their amino acid sequences and AT-hook and PPC domain, the *LcAHL* genes were divided into two phylogenetic clades. We investigated the cis-acting elements of the promoter regions and found that the promoters of *AHL* possess drought stress-related *cis*-regulatory elements. After that, we systematically studied the *LcAHLs* expression profiles in different tissues and the response to stress conditions, as well as the expression at different stages of somatic embryogenesis. Our study found that the *LcAHL* genes show functions on drought stress resistance and take part in somatic embryogenesis. RNA-seq analysis showed that the expression of *LcAHLs* increases during somatic embryo formation, especially in torpedo-shaped embryos and mature cotyledon embryos. Transcriptome data of tissue and drought stress showed that *LcAHLs* are mainly expressed in floral organs and that their expression level increased under drought stress. These results laid a molecular foundation for improving the efficiency of somatic embryos and cultivating drought resistant plants. It provides alternative genes for *L. chinense* to better adapt to the arid natural environment and improve its somatic embryogenesis efficiency.

## Figures and Tables

**Figure 1 plants-12-01353-f001:**
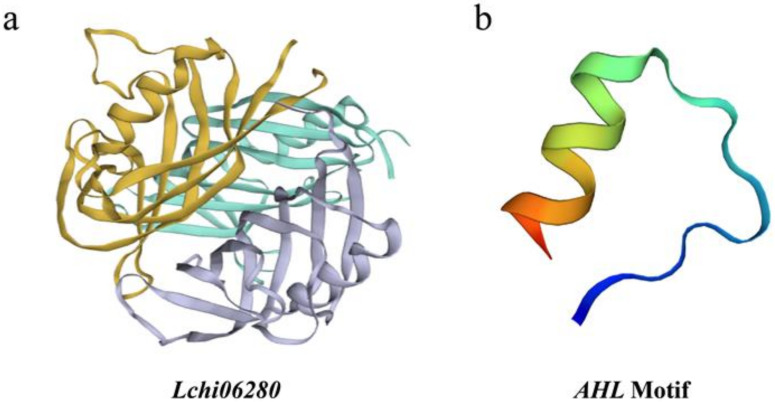
The tertiary structure of Lchi06280 protein. (**a**) The complete protein tertiary structure of Lchi06280. Different colors represent different structures. (**b**) The conserved domain of Lchi06280.

**Figure 2 plants-12-01353-f002:**
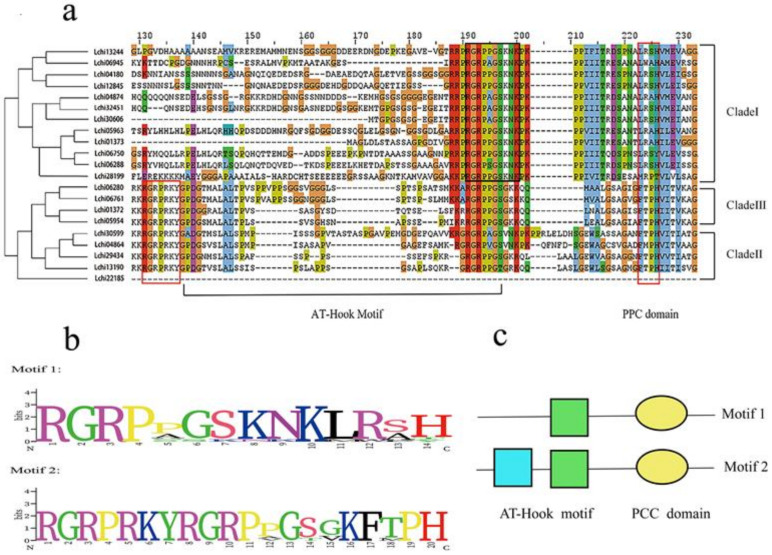
Multiple sequence alignment analysis results of LcAHLs. (**a**) The obtained phylogenetic tree is shown on the left, and the conserved domain in two functional units of the AHL proteins, the AT-hook motif and PPC domains, is shown on the right. (**b**) Sequence information of conserved domain proteins in different branches. The *x*-axis and *y*-axis suggest the conserved sequences of the domain and the conservation rate of each amino acid, respectively. (**c**) The topology of two LcAHLs types (type I and type II) was consistent with the PCC domain and AT-hook motifs.

**Figure 3 plants-12-01353-f003:**
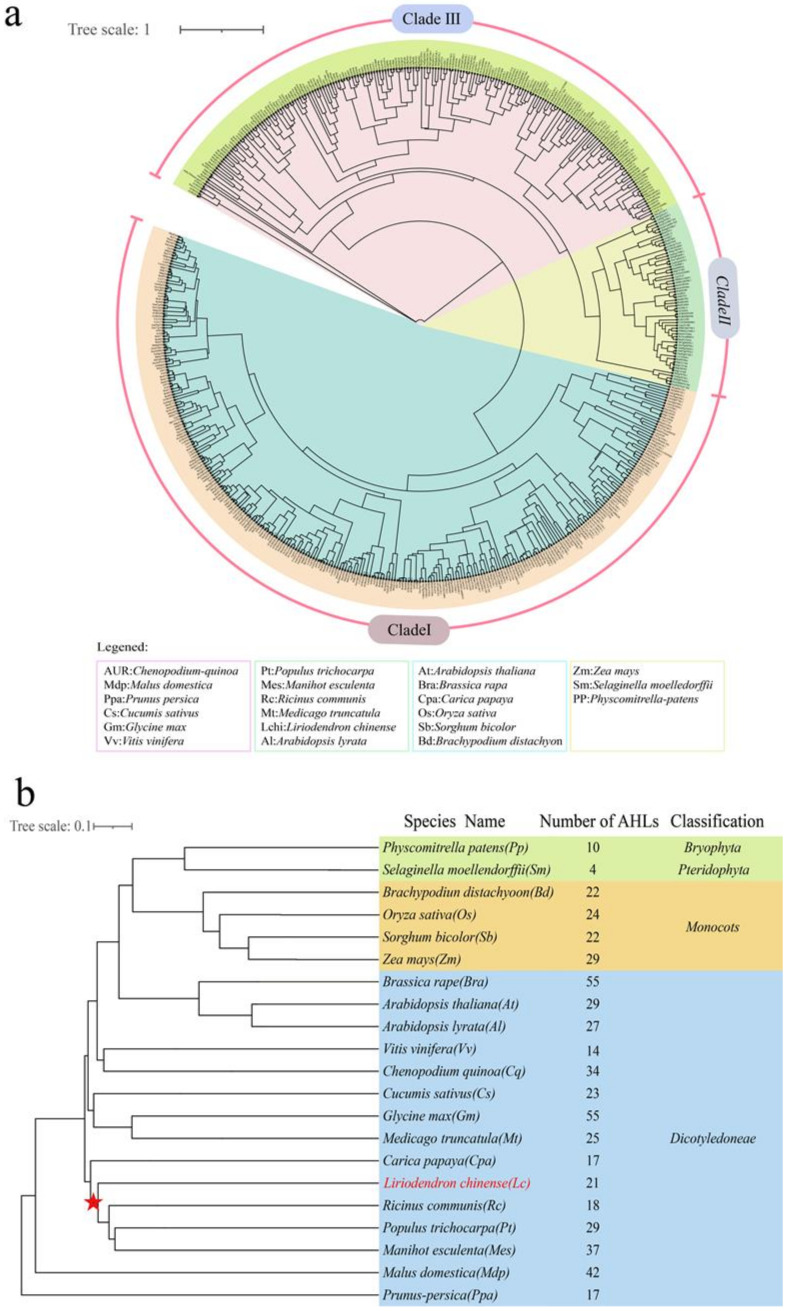
Phylogenetic analysis and species tree of the AHL proteins in 21 species. (**a**) Different colors indicate different evolutionary clades. The Bayesian tree shows the phylogenetic relationships between AHL proteins. The tree is divided into three branches, including Clade I, Clade II, and Clade III. (**b**) The green highlight is classified as Bryophyta and Pteridophyte, the orange highlight is classified as the monocots, and the blue highlight is classified as Dicotyledonae. Red stars represent the branching sites of *L. chinense*, and the red font represents the evolutionary position of *L. chinense*.

**Figure 4 plants-12-01353-f004:**
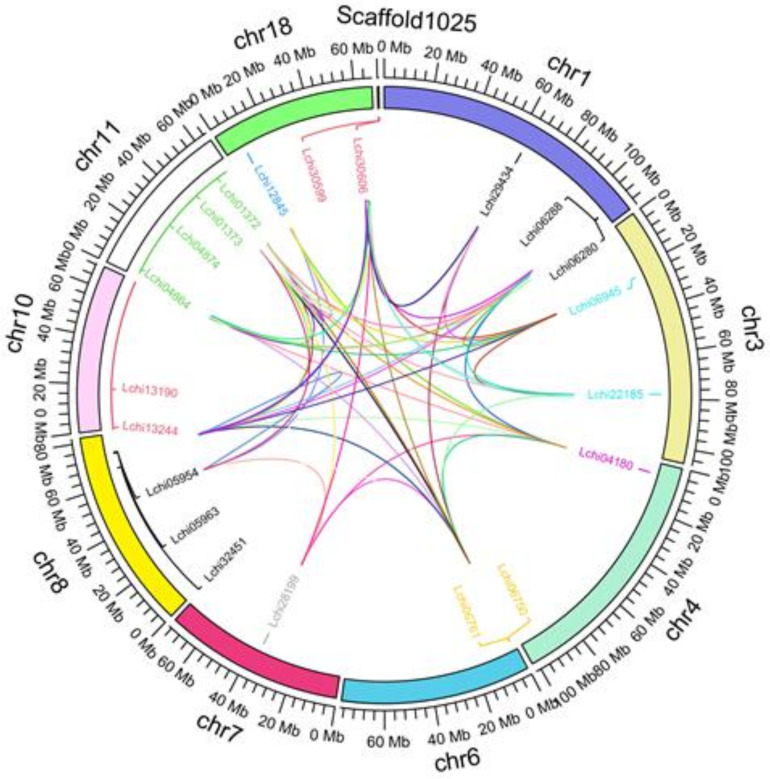
Chromosome mapping and gene replication event analysis of 21 LcAHLs. Chromosomes are represented by different colors, and the outer circle represents the scale. The inner circle is the location information of the gene. The arcs of different colors represent gene pairs of different gene replication events.

**Figure 5 plants-12-01353-f005:**
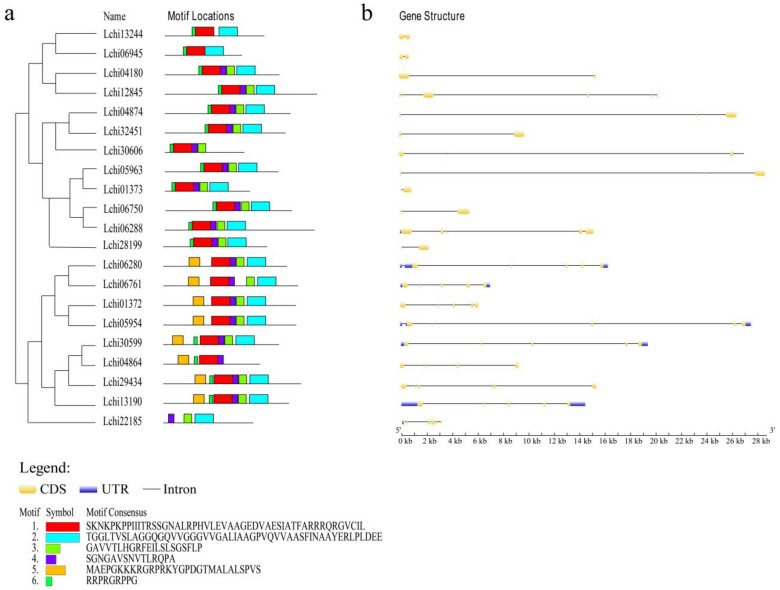
Conserved motif and gene structure information of LcAHLs. (**a**) Conserved motif information of 21 LcAHLs. We predicted 6 conserved motifs of LcAHL proteins and have marked them with different colors. (**b**) The *x*-axis represents the predicted length of different *LcAHL* genes (5′ to 3′). Different colors represent different structures. Yellow rectangles are CDS, blue rectangles are UTRs, and the black straight line represents introns. The scale represents gene length (bp).

**Figure 6 plants-12-01353-f006:**
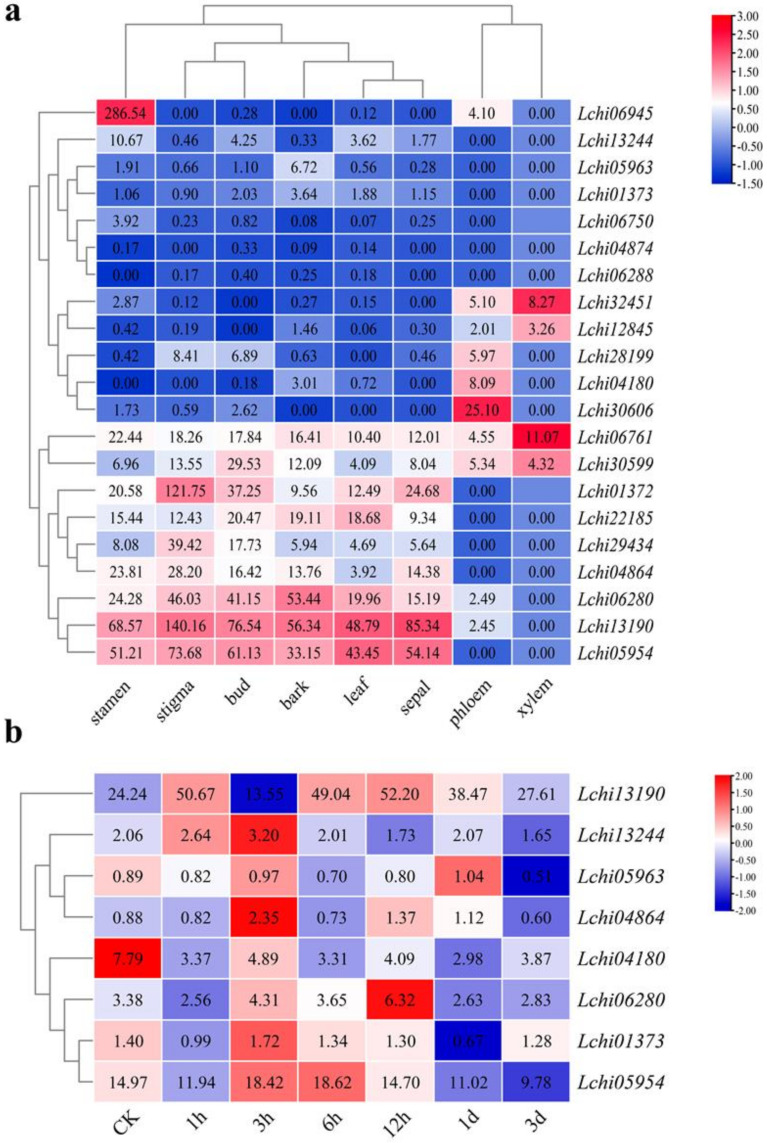
Expression pattern of *LcAHL* genes in eight tissues and under drought stress. (**a**) Heatmap of *LcAHL* genes in eight tissues. The color scale represents the values of log2 fold change, red represents a high level, and blue indicates a low level of transcript abundance. The heatmap includes phloem, stigma, xylem, bud, stamen, leaf, bark, and sepal. We performed row clustering and column clustering so that similar expression patterns were clustered together. (**b**) Heatmap of 8 *LcAHL* genes under drought stress. We performed row clustering so that genes with the same expression trend were clustered together.

**Figure 7 plants-12-01353-f007:**
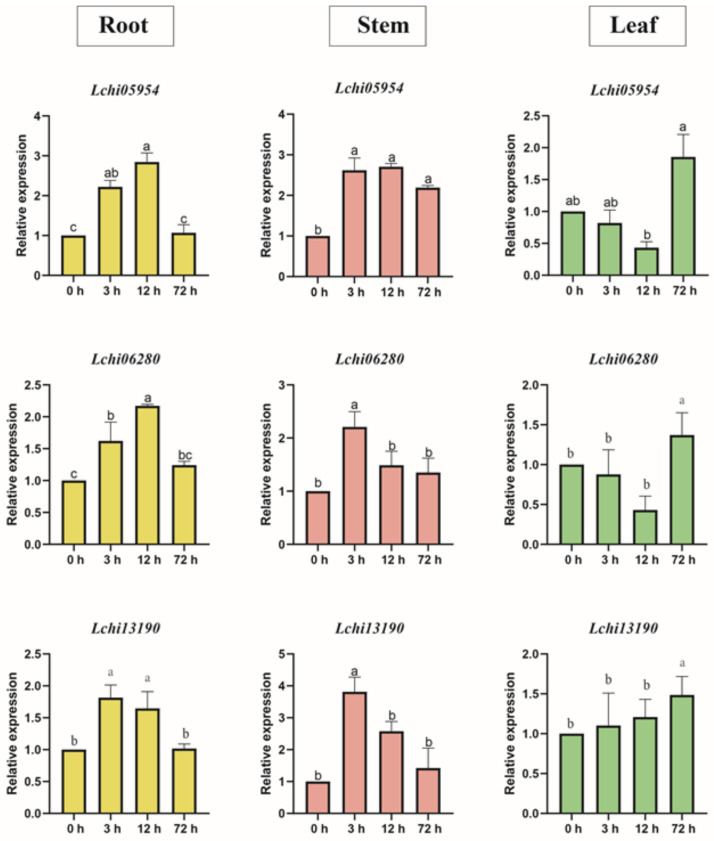
qRT-PCR verification of *LcAHLs* expression in different tissues under drought stress. Different colors of the pillars represent different tissues. A histogram in the same row represents different tissues of the same gene. Error bars represent the deviations from three biological replicates. Mark the letter a on the maximum average and marker b with significant difference, otherwise marker a (α = 0.05), and so on.

**Figure 8 plants-12-01353-f008:**
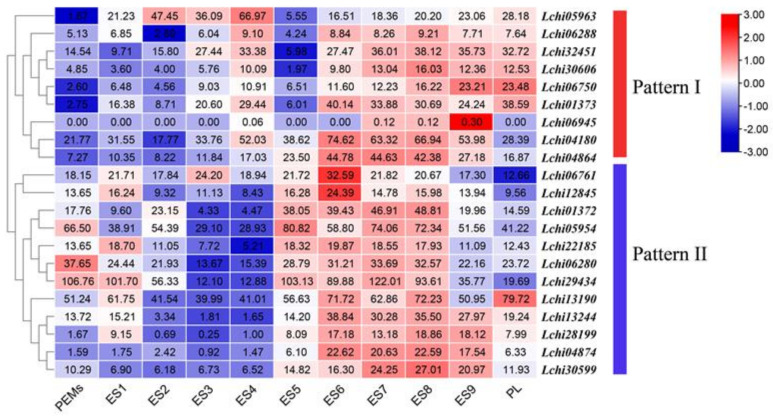
Expression profiles of 21 *LcAHL* genes in somatic embryogenesis. The color scale represents the values of log2 fold change, red represents a high level, and blue indicates a low level of transcript abundance. Similar expression patterns are clustered together.

**Figure 9 plants-12-01353-f009:**
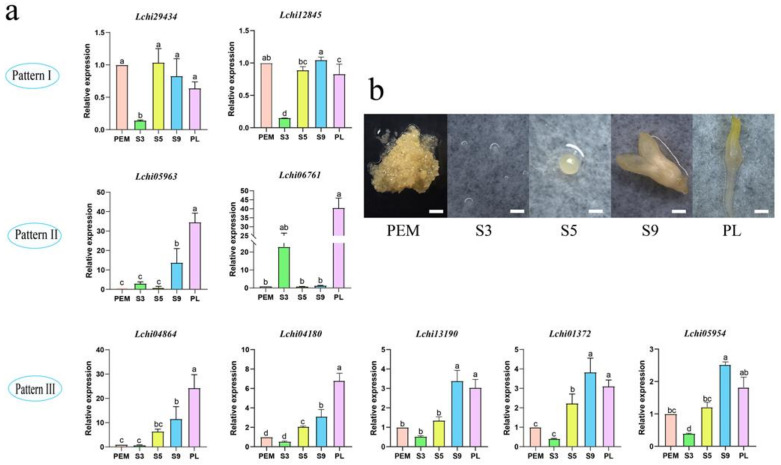
qRT-PCR verification of *LcAHL* gene expression at different stages during somatic embryogenesis. (**a**) qRT-PCR result of *AHLs* during somatic embryogenesis. The *y*-axis shows the relative expression level, and the *x*-axis indicates different stages of somatic embryogenesis. At the same time, three biological and three technical replicates per period are shown. Error bars represent the deviations from three biological replicates. Mark the letter a on the maximum average and marker b with significant difference, otherwise marker a (α = 0.05), and so on. (**b**) Different sampling periods of somatic embryogenesis in *Liriodendron chinense*. The scale represents 1 mm. PEMs: proembryogenic mass cultured for 20 days for embryogenic callus, ES3: induced by ABA for 1 day, ES5: 7-day globular embryo, ES9: 31-day mature cotyledon embryo.

**Figure 10 plants-12-01353-f010:**
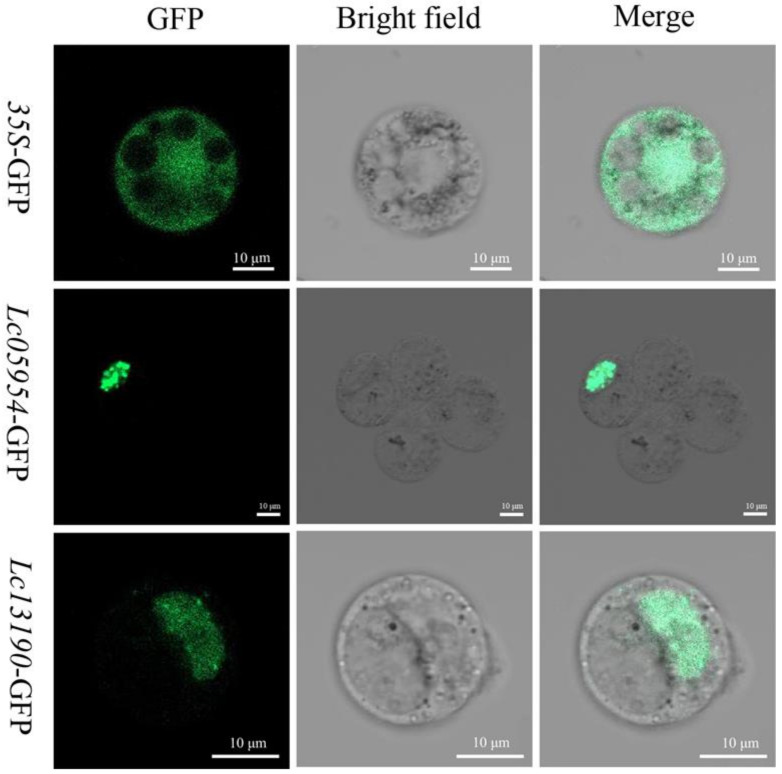
Subcellular location of LcAHL proteins. The scale represents 10 μm.

**Table 1 plants-12-01353-t001:** Physicochemical properties of LcAHLs.

Gene ID	pI	MW(kDa)	Subcellular Localization	GRAVY	Instability Index (II)	Clade
*Lchi01372*	9.54	37.35	Nucleus	−0.450	45.15	III
*Lchi05954*	9.75	37.14	Nucleus	−0.448	52.16	III
*Lchi06280*	10.25	33.71	Nucleus	−0.372	57.95	III
*Lchi06761*	10.62	36.90	Nucleus	−0.191	57.51	III
*Lchi04864*	8.53	27.28	Nucleus	−0.367	41.69	II
*Lchi30599*	9.48	71.46	Nucleus	−0.281	44.51	II
*Lchi29434*	9.24	39.47	Nucleus	−0.234	55.52	II
*Lchi13190*	10.14	34.91	Nucleus	−0.387	49.51	II
*Lchi22185*	6.95	25.80	Chloroplast	0.010	61.85	II
			Mitochondrion			
*Lchi01373*	8.55	23.58	Nucleus	0.055	50.15	I
*Lchi05963*	6.38	32.57	Nucleus	−0.148	53.22	I
*Lchi06288*	6.43	72.82	Chloroplast	−0.107	49.43	I
*Lchi06750*	4.71	36.33	Nucleus	−0.551	71.02	I
*Lchi28199*	8.88	29.33	Chloroplast	−0.190	61.53	I
			Nucleus			
*Lchi04874*	6.25	36.05	Nucleus	−0.547	55.18	I
*Lchi32451*	5.85	34.12	Nucleus	−0.427	44.06	I
*Lchi30606*	10.00	23.17	Chloroplast	−0.446	42.67	I
*Lchi04180*	10.25	33.71	Mitochondrion	−0.166	50.88	I
*Lchi12845*	8.23	43.19	Chloroplast	−0.288	45.90	I
*Lchi06945*	9.57	22.38	Nucleus	−0.493	40.80	I
*Lchi13244*	5.98	27.41	Nucleus	−0.487	48.11	I

Note: pI: isoelectric point; MW: molecular weight; GRAVY: grand average of hydropathicity.

**Table 2 plants-12-01353-t002:** The length and position information of LcAHL.

ID	Gene Length	Pos. (Chr)	CDs Length	Exon Number	Protein Length (aa)
*Lchi01372*	6154	Chr11	1083	5	270
*Lchi01373*	830	Chr11	696	2	163
*Lchi04180*	15,499	Chr4	936	2	219
*Lchi04864*	9361	Chr11	792	4	201
*Lchi04874*	26,530	Chr11	1026	3	229
*Lchi05954*	27,582	Chr8	1086	5	275
*Lchi05963*	28,703	Chr8	930	3	207
*Lchi06280*	16,339	Chr1	1011	5	248
*Lchi06288*	15,233	Chr1	2028	5	474
*Lchi06750*	5428	Chr6	1035	3	232
*Lchi06761*	7049	Chr6	1101	5	278
*Lchi06945*	762	Chr3	630	2	158
*Lchi12845*	20,351	Chr18	1242	6	305
*Lchi13190*	14,467	Chr10	1026	5	256
*Lchi13244*	904	Chr10	813	2	203
*Lchi22185*	3109	Chr3	735	6	187
*Lchi28199*	2170	Chr7	849	2	212
*Lchi29434*	15,391	Chr1	1125	5	269
*Lchi30599*	19,424	Scaffold1025	945	5	329
*Lchi30606*	27,410	Scaffold1025	651	3	151
*Lchi32451*	9846	Chr8	987	2	225

**Table 3 plants-12-01353-t003:** *Cis-acting* element quantification and function prediction.

*Cis*-Acting Element	Number	Function
GARE-motif	14	gibberellin-responsive element
TCA-element	9	salicylic acid responsiveness
CGTCA-motif	19	MeJA-responsiveness
MBSI	15	flavonoid biosynthetic genes regulation
O2-site	8	zein metabolism regulation
ABRE	56	abscisic acid responsiveness
AuxR-core	8	auxin responsiveness auxin responsiveness
TGA-element	4	
GCN4_motif	6	endosperm expression
MSA-like	2	cell cycle regulation
LTR	25	low-temperature responsiveness
TC-rich repeats	5	defense and stress responsiveness
MBS	15	drought-inducibility

## Data Availability

Transcriptome data of somatic embryogenesis and tissues have not yet been published. The drought stress transcriptome data of *L. hybrid* is annotated with accession number PRJNA679101 and can be downloaded through NCBI (https://www.ncbi.nlm.nih.gov/bioproject/PRJNA679101/, accessed on 5 September 2022). The complete genome, transcript/protein sequences, and genome feature file of *Lchi* were downloaded from https://www.ncbi.nlm.nih.gov/assembly/GCA_003013855.2, accessed on 5 September 2022.

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
