# Peer review of "AT-Hook Transcription Factors Show Functions in Liriodendron chinense under Drought Stress and Somatic Embryogenesis"

_plants, 2023, doi:10.3390/plants12061353_

Round 1

Reviewer 1 Report

the paper is very important and good

add space between the number of references and previous words

try to increase the resolution of figure 3 and the font of the words

in figure 7 some histogram not contains statistical analysis please add it

Reviewer 2 Report

It is established rule in scientific writing that introduction should be representative of title. For any systematic elaboration of introduction, flow must follow the trend of the title.1.         What’s the gap of knowledge? Which is the scope of the manuscript? What hypothesis have been made? The introduction should be revised accordingly.

2. After that, give a practical problem which is identified and the aims of the study.
3-     Give a clear-cut statement about the aims/objectives of the study at the end of the introduction.
4-     Also, give the target audience who will get benefit from this.
5-     Novelty statement is missing. Please provide a statement showing the significance of your study compared to other documented literature.

6-     The objectives must be SMART, i.e., (S - Specific, M - Measurable, A - Achievable, R - Realistic, T – Time-bound). Please rephrased the objective statements because no SMART characteristics in objectives are present. 

7- The scientific background of the topic is poor. In "Introduction" and "Discussion", the authors should cite recent references between 2016-2022 from JCR journals.

Ali S, Hameed G, Muhammad A, Depeng W, Fahad S (2022) Comparative Genetic Evaluation of Maize Inbred Lines at Seedling and Maturity Stages Under Drought Stress. J Plant Growth Regul https://doi.org/10.1007/s00344-022-10608-2

Fahad S, Bajwa AA, Nazir U, Anjum SA, Farooq A, Zohaib A, Sadia S, NasimW, Adkins S, Saud S, Ihsan MZ, Alharby H,Wu C,Wang D, Huang J (2017) Crop production under drought and heat stress: Plant responses and Management Options. Front Plant Sci 8:1147. https://doi.org/10.3389/fpls.2017.01147

Reviewer 3 Report

Dear Authors,

In my opinion, the manuscript "AT-hook transcription factors show functions in Liriodendron chinense under drought stress and somatic embryogenesis", presented for my assessment is generally written in the correct form. The information contained in it could be cognitive and application significant and I think the presented data and conclusions could interest many researchers and readers after some refilling. Although the work is interesting but presented in such an approach is not convincing and does not exhaust the topic. I think that the Authors should take into account a thorough modification of this article. I recommend publishing it in "Plants” after a minor revision.

General comments:

I would like to make general remarks here:

1.      Authors should review the manuscript very carefully in editorial context, e.g. notoriously lack of spaces when citing articles in the manuscript text, it should be corrected.

2.      The Authors should clarify more the goals of the research undertaken.

Several specific comments:

Abstract

The abstract is too wordy, in my opinion, it should be shortened, containing a brief background of the research, information on the methodology used, and the most important results.

Moreover, the abstract does not correlate with the chapter "Conclusions", it should be corrected

Keywords

Authors should take into account that keywords, according to the rules of writing scientific papers, should not be the same as in the title and usually in alphabetical order.

Introduction

The use of species names should be standardized, preferably scientific, Latin names, while common names are mixed with Latin names (Line 41 – 47)

Line 42: in my opinion, there is a lack of citations at the end of the sentence

Line 70 – 74: there is a lack of proper definition of somatic embryogenesis, it should be corrected

Importantly, there is no information about the use of the somatic embryogenesis process.

It is known that in the experimental setup, it occurs mainly in in vitro culture and that there are many analogies in the course of SE and zygotic embryogenesis (ZE). Whether the authors wanted to address these similarities is not known. So you need to be more clear about your goals. The authors have performed many molecular analyzes, thus bringing the process of somatic embryogenesis closer, but what is the main goal, or just cognitive in itself, can only be guessed.

Moreover, how to bring together the aspect of drought and the process of somatic embryogenesis is difficult to read from this work. It's worth explaining

Results

It would be good to include information on what stage of ontogenesis the research was conducted, I did not find such information

Line 228 – 230: “1h after drought stress induction, the expression of Lchi13190 and 228 Lchi13244 increases, while Lchi05963 and Lchi04864 tend to remain stable, and Lchi04180, 229 Lchi06280, Lchi01373, and Lchi05954 decrease their expression” - language awkwardness, should be corrected

Line 292 – 303: information on the methodology of the experiment should be found in Materials and Methods, not here – must be corrected.

Conclusions

I advise you to emphasize the conclusions more, and as noted earlier, this part should be consistent with the abstract part.

With best regards!

Round 2

Reviewer 2 Report

Accepted as it stands